# *Origanum majorana* L. as Flavoring Agent: Impact on Quality Indices, Stability, and Volatile and Phenolic Profiles of Extra Virgin Olive Oil (EVOO)

**DOI:** 10.3390/foods13193164

**Published:** 2024-10-04

**Authors:** Panagiota Kyriaki Revelou, Spyridon J. Konteles, Anthimia Batrinou, Marinos Xagoraris, Petros A. Tarantilis, Irini F. Strati

**Affiliations:** 1Laboratory of Chemistry, Analysis & Design of Food Processes, Department of Food Science and Technology, University of West Attica, Agiou Spyridonos, 12243 Egaleo, Greece; p.revelou@uniwa.gr (P.K.R.); skonteles@uniwa.gr (S.J.K.); batrinou@uniwa.gr (A.B.); 2Laboratory of Chemistry, Department of Food Science and Human Nutrition, Agricultural University of Athens, Iera Odos 75, 11855 Athens, Greece; mxagor@aua.gr (M.X.); ptara@aua.gr (P.A.T.)

**Keywords:** flavored extra virgin olive oil, *Origanum majorana* L., quality indices, oxidative stability, chlorophyll and *b*-carotene content, antifungal activity, volatile profile, phenolic profile

## Abstract

The flavoring of olive oils with aromatic plants is commonly used to enrich the oils with aromatic and antioxidant compounds. *Origanum majorana* L. was applied as a flavoring agent for extra virgin olive oil (EVOO), at concentrations of 20 g L^−1^ and 40 g L^−1^, via ultrasound-assisted maceration. The aim of this study was to evaluate the impact of flavoring on the EVOOs’ quality indices, oxidative stability, and antioxidant, antiradical and antifungal activities, as well as on the oils’ volatile and phenolic profile. The flavored EVOO maintained the quality indices (free fatty acids, peroxide value, extinction coefficients) below the maximum permitted levels, whereas the addition of marjoram enhanced the oxidative stability, the levels of chlorophyll and b-carotene and the total phenolic content. The incorporation of marjoram into the EVOO did not have a significant impact on the antioxidant and antiradical activities. Concerning the antifungal activity, no *Zygosaccharomyces bailli* cell growth was observed for two weeks in a mayonnaise prepared with the flavored EVOO at a 40 g L^−1^ concentration. SPME-GC-MS analysis revealed the presence of 11 terpene compounds (hydrocarbon and oxygenated monoterpenes) that had migrated from marjoram in the flavored EVOO. Twenty-one phenolic compounds were tentatively characterized by LC-QToF-MS in the EVOO samples; however, hesperetin and p-coumaric acid, originating from marjoram, were only detected in the flavored EVOO.

## 1. Introduction

The Mediterranean diet has gained widespread popularity all around the globe due to the recognized positive effects on consumers’ well-being and health [1]. Olive oil plays a key role in Mediterranean cuisine, and numerous studies have highlighted its cancer-preventive and cardiovascular benefits, largely due to its rich phenolic compounds and high levels of monounsaturated fatty acids [2,3]. However, to provide effective health benefits, olive oil must include specific bioactive compounds, such as phenolics, as outlined in the polyphenols-related health claim [4].

Olive oil is often flavored to enhance its sensory appeal, health benefits and shelf life. The flavoring of olive oil can be performed by adding dry herbs, essential oils, or plant extracts or by co-extraction [5,6]. The oxidative stability of olive oil, which affects its shelf life, can be promoted by the addition of antioxidants, which are obtained from the flavoring agents. On the other hand, studies have shown that the addition of various flavorings can lead to the appearance and survival of certain microorganisms (including molds, yeasts and bacteria), depending on the concentration and aromatizer used [7]; however, a great challenge remains concerning the use of flavorings as natural antifungal agents in the replacement of benzoates and sorbates, which are widely used in foods susceptible to yeasts and fungi.

It has been suggested that olive oil’s health benefits are improved by the addition of flavorings derived from aromatic plants. For example, the volatile compounds (VCs) found in the plants of the *Lamiaceae* family are known for their antioxidant properties and nutritional benefits [8,9]. A number of studies have focused on olive oils flavored with *Lamiaceae* plants, including basil [10,11], rosemary [10,11,12], lavender [11], mint [11], oregano [12,13], sage [11], and thyme [12,14]. *Origanum majorana* L.—sweet marjoram—in particular, has been the subject of researchers’ interest. The essential oil derived from the plant has been found to contain terpenes and phenolic compounds with notable antioxidant [15,16], anti-inflammatory [17], anticancer [15], hepatoprotective [18], and anti-acetylcholinesterase properties [19].

Ultrasound and microwave technology were successfully used to improve the flavoring process [20,21]. Ultrasound is particularly promising because of its green credentials, and it has been successfully used to flavor olive oil with rosemary and basil [10], black pepper [13], caraway [22], olive tree leaves [23], thyme [24], chili peppers [25], and pink pepper [26].

While a number of studies have been carried out on olive oils flavored with plants from the *Lamiaceae* family, to date, there appear to be no published studies on olive oil flavored with *Origanum majorana* L. The aim of the current research was (a) the use of *Origanum majorana* L. as a flavoring agent for extra virgin olive oil (EVOO) with the aid of ultrasound technology and (b) the impact of flavoring on the EVOO’s quality indices, oxidative stability, antifungal activity, as well as on its volatile and phenolic profiles.

## 2. Materials and Methods

### 2.1. Study Material—Preparation of Flavored Olive Oil

The extra virgin olive oil (EVOO) used in this study was produced during the period November–January 2023 from single-varietal olives of Greek Koroneiki variety, cultivated in south Peloponnese. The olive oil was processed within 24 h, using the cold-pressing technique, from a two-phase decanter. The oil was stored in dark glass containers at 15 °C until further treatment and analysis. The aromatic plant marjoram (*Origanum majorana* L.) was cultivated in the area of Megali Chora (Municipality of Agrinio of the Prefecture of Aetoloakarnania, Western Greece) and was collected by the company ANTHIR A.B.E.E. (Agrinion, Greece), which specializes in the production and processing of organic medicinal and aromatic plants (MAPs) in the form of dry herbs, extracts and essential oils.

For the preparation of the flavored EVOO samples, two samples, approximately 10.0 g (sample A) and 20.0 g of dry marjoram (sample B), were weighted and added to 500 mL of extra virgin olive oil in two 600 mL glass bottles. The bottles were subsequently put in an ultrasonic laboratory bath DU-65 (Argolab, Carpi, Italy) and the ultrasounds were applied at level 5 (P = 180 W) for 15 min. The temperature was set at 20 °C. After 15 min, samples A and B were filtered and the filtrates were divided into two groups. The first group (samples coded A_0 and B_0) was analyzed immediately (0 day of storage), whilst the second group was stored at 23–25 °C in a dark place and analyzed after 14 days of storage (samples coded A_14 and B_14). Similarly, a sample of extra virgin olive oil, without the addition of dry marjoram (control sample C), was prepared and subjected to the same ultrasonic maceration process as mentioned above. The whole procedure for the EVOO flavoring was repeated twice.

### 2.2. Determination of Olive Oil Quality Indices

The evaluation of the established quality indices for olive oil—specifically, the free fatty acids, peroxide value, and specific UV absorption constants (K232, K270, and ΔK)—was carried out using the official methods outlined by the International Olive Council, according to the codes COI/T.20/Doc. No 34/Rev. 1 2017, N°35/Rev. 1 2017, and N°19/Rev. 5 2019, respectively [27].

### 2.3. Color Measurement

The color of the oil samples was analyzed in glass cells using a Lovibond^®^ Model Fx Spectrocolorimeter (The Tintometer Limited, Lovibond House, Sun Rise Way, Amesbury, SP4 7GR, UK). This instrument provided color measurements based on the Lovibond^®^ RYBN, chlorophyll, and β-carotene scales, with the color units expressed as neutral (N), red (R), yellow (Y), and blue (B), following the AOCS Official Method Cc 13e-92 [28]. The Lovibond scale relies on the color absorbance of various glasses, determining the Lovibond degrees by comparing the color of light transmitted through the sample with that of light passing through a series of standard colored glass filters. The combination of primary colors (red, yellow, blue) and neutral filters that match the sample’s absorption defines the RYBN Lovibond color.

### 2.4. Determination of Oxidative Stability

The oxidative stability of the flavored extra virgin olive oils was determined by the Rancimat method, using the modified AOCS Official Method Cd 12b-92 (2017c) [29]. The Rancimat method evaluates the stability by measuring the oxidation induction time (OIT) using the Rancimat apparatus (Metrohm 743, Metrohm, Herisau, Switzerland), which is capable of operating over a temperature range of 50 to 220 °C. The oil samples (3 ± 0.2 g) were placed in Rancimat vessels with an air flow rate of 20 L/h passing through them, and afterwards, they were put in an electric heating block at 120 °C. The effluent air containing the volatile secondary products of lipid oxidation from the oil samples was collected in measuring vessels filled with 60 mL of deionized water. The resulting increase in the water conductivity was measured and plotted against the time. The OIT (in hours), also defined as the OSI (oxidative stability index), is the time taken until there is a sharp increase in conductivity, determined as the point of inflection on the conductivity versus time curve.

### 2.5. Determination of Antifungal Activity

In order to investigate the antifungal activity of the flavored EVOO, mayonnaise, a characteristic oil in water emulsion with high acidity and osmotic pressure, was selected as the model system. The target microorganism was *Zygosaccharomyces bailii*, a yeast highly resistant to environmental stress that frequently causes spoilage in dressings like mayonnaise, juices, and soft drinks [30].

#### 2.5.1. Inoculum Preparation

The *Zygosaccharomyces bailli* DSMZ 70834 stock culture was stored at −80 °C by being frozen in 20% *v*/*v* glycerol and 80% yeast peptone dextrose (YPD, Sigma Aldrich, Burlington, MA, USA) broth medium. A loopful of the frozen culture was transferred into 10 mL YPD broth and incubated at 25 °C for 48 h. From the broth culture, 10 μL was transferred into 10 mL YPD broth and incubated under the same conditions. Following the second incubation, the broth culture was centrifuged under aseptic conditions (7500 g/10 min), and the biomass was suspended in peptone water (PW, HIMEDIA, Modauta, Germany). Then, serial ten-fold dilutions in PW were carried out, and the optical density (OD 615 nm) of each dilution was measured. Additionally, the yeast cell count of each dilution was counted on YPD after incubation of the plates at 25 °C for 48 h in order to determine the inoculum concentration. In this way, the inoculum population for the three replicates of the mayonnaise challenge test were nearly identical (approximately 10^4^ cfu/mL).

#### 2.5.2. Challenge Test

Microbiological challenge testing is a useful tool for determining the ability of a food to support the growth of spoilage organisms or pathogens. During this test, a deliberate inoculation of the product with the test microorganism is performed and the product is then stored and tested for this microorganism during its shelf life [31].

A batch of mayonnaise (MY B_0), 250 g, was prepared under laboratory conditions by mixing the following ingredients: 2 egg yolks, 1 tablespoon mustard (15 g), 1 tablespoon white vinegar (15 g), 5 g salt, and 240 mL EVOO flavored with 40 g L^−1^ dry marjoram (similar to sample B described in Section 2.1). The final pH was 4.0. A second batch of mayonnaise (MY C) was prepared accordingly as a control by replacing the flavored with unflavored EVOO. Each batch was transferred into a stomacher bag and inoculated with 20 mL of inoculum of the spoilage yeast *Zygosaccharomyces bailii* (10^4^ cfu/mL), followed by mixing in a stomacher for 1 min for an even distribution of yeast cells. Each batch was apportioned into sterile jars, each containing 15 g of mayonnaise, which were then placed in an incubator at 25 °C for a total period of 20 days. The yeast counts were performed at 0, 5, 10, 15 and 20 days, in triplicate, under the following procedure: ten grams of mayonnaise was mixed under aseptic conditions in a stomacher bag with 90 mL of PW. This was followed by serial decimal dilutions and then spread-plated on YPD plates that were incubated at 25 °C for 48 h. The colony forming units (CFU) were counted and were expressed as the log CFU/g. Each sampling day, the pH was measured.

### 2.6. Identification and Semi-Quantification of Volatile Compounds (VCs) by Solid Phase-Microextraction Gas Chromatography-Mass Spectrometry (SPME-GC-MS)

The analysis of the VCs was performed according to the method of Revelou et al. (2021) [32]. Five grams of olive oil was put into a 15 mL vial along with 1 μL of internal standard (β-ionone, Alfa Aesar, Ward Hill, MA, USA). The olive oil sample was heated at 50 °C in a water bath and stirred at 700 rpm for 30 min.

A divinylbenzene/carboxen/polydimethylsiloxane (DVB/CAR/PDMS) fiber (Supelco, Bellefonte, PA, USA) was inserted and exposed to the headspace of the olive oil sample for 15 min. After the completion of sampling, the fiber was inserted into the injector of the gas chromatograph.

Analysis of the VCs was performed using a gas chromatograph Thermo GC-TRACE ultra, coupled to a Thermo Mass Spectrometer DSQ II (Thermo Scientific Inc., Waltham, MA, USA) with a Restek Rtx-5MS, 30 m × 0.25 mm × 0.25 μm (Restek, Bellefonte, PA, USA) column. The helium carrier gas flow rate was 1.0 mL/min. The GC inlet was operated for 3 min in splitless mode at a temperature of 260 °C. The column temperature was held for 6 min at 40 °C, then increased to 120 °C with a rate of 5 °C/min, the temperature increased to 160 °C with a rate of 3 °C min, heated to 250 °C with a rate 15 °C/min and held at 250 °C for 1 min. The temperatures of the MS source, quadrupole and transfer line were 240 °C, 150 °C and 290 °C, respectively. The MS operated at a mass range of 35–650 *m*/*z* with an acquisition mode of electron impact 70 eV.

The VCs were identified by the comparison of the spectral data and the retention index (RI) of the eluting compounds to those of the Adams [33] and Wiley 275 mass spectra library. An n-alkane (C8–C20) standards solution (Supelco, Bellefonte, PA, USA) was used for the RI calculation. Semi-quantification of the VCs was performed by dividing the peak area of each compound by the peak area of the internal standard. This ratio was multiplied by the initial concentration of β-ionone and expressed as mg kg^−1^.

### 2.7. Extraction of Phenolic Compounds

The phenolic extracts from the EVOO were prepared according to the International Olive Council method [34] Briefly, 5.0 mL of methanol/water 80/20 (*v*/*v*) solution was mixed with 2.0 g of olive oil and the resulting emulsion was stirred vigorously for 1 min with a vortex. Subsequently, the mixture was extracted in the ultrasonic bath for 15 min at room temperature. After extraction, the samples were centrifuged at 5000 rev/min for 25 min and the supernatants were collected and filtered through a 5 mL plastic syringe with a 0.45 μm PVDF filter. The phenolic extracts were collected in tubes covered with aluminum foil and were preserved at −20 °C until further analyses.

### 2.8. Spectrophotometric Assays

The total phenolic content (TPC) in the olive oil samples was measured using a modified Folin–Ciocalteu assay. The measurements were performed in triplicate with a Spectro 23 Digital Spectrophotometer (Labomed, Inc., Los Angeles, CA, USA), and the absorbance was recorded at 750 nm. The results were expressed as milligrams of gallic acid equivalents (GAE) per kilogram of olive oil, with a calibration curve generated from standard solutions of gallic acid ranging from 20 to 500 mg/L [35].

To evaluate the antiradical activity of the olive oil samples against the ABTS•+ radical, the method outlined byLantzouraki et al. (2015) [36] was utilized. The absorbance was measured at 734 nm, and the antiradical activity was reported as milligrams of Trolox equivalents (TE) per kilogram of olive oil. Calibration curves were created using Trolox standard solutions ranging from 0.20 to 1.5 mM.

The ferric-reducing antioxidant power (FRAP) assay, based on the protocol by Lantzouraki et al. (2016) [37], was conducted to assess the antioxidant activity. Absorbance readings were taken at 595 nm, with the results expressed as milligrams of Fe(II) equivalents per kilogram of olive oil. A calibration curve was constructed using standard solutions of FeSO_4_ × 7H_2_O at concentrations from 600 to 2000 μM.

### 2.9. Identification of Phenolic Compounds by High-Resolution Liquid Chromatography-Mass Spectrometry

An Agilent 6530 Quadrupole Time of Flight (QToF) mass spectrometer coupled with a UHPLC system was used to obtain the mass spectra (MS) and the MS/MS spectra in negative electrospray ionization (ESI) mode (Agilent Technologies, Santa Clara, CA, USA), according to a previously published method [38].

For the liquid chromatography analysis, a Nucleoshell EC C18 column was used (100 mm × 4.6 mm, 2.7 μm) (Macherey-Nagel GmbH & Co., Düren, Germany). The mobile phase consisted of (A) ultrapure water–acetic acid 0.1% and, (B) acetonitrile–acetic acid 0.1%. Acetonitrile and acetic acid of LC-MS grade were purchased from Merk KGaA (Darmstadt, Germany). A Genie Water System was utilized to obtain the ultrapure water (RephiLe Bioscience Ltd., Shanghai, China). The gradient program was as follows: 0 min: 10% B; 8 min: 30% B; 12 min: 40% B; 16 min: 50% B; 18 min: 10% B; 33 min: 10% B. A flow rate of 1.0 mL min^−1^ was applied, with an injection volume of 10 μL. The chromatographic analysis was performed at 30 °C. The acquisition and processing of the data were performed utilizing the Agilent MassHunter software (versions B.06.00 and B.07.00, respectively).

Phenolic compounds were identified by comparison of the retention time of the standards (p-coumaric acid, luteolin, naringenin, apigenin, and hesperetin, and vanillic acid), the MS/MS spectra and the exact mass. The standards were obtained from Extrasynthese (Genay, France). The Riken database (http://spectra.psc.riken.jp/) and the human metabolome (HMDB) database (https://hmdb.ca/) were used to identify compounds for which no standards were available.

### 2.10. Statistical Analysis

ANOVA and post hoc comparisons were performed by applying the Tukey HSD test using the SPSS software (version 28.0, IBM Corp., Armonk, NY, USA).

## 3. Results and Discussion

### 3.1. Extra Virgin Olive Oil Quality Indices and Lovibond Color Parameters

Table 1 presents the quality indices, the oxidation induction time and the Lovibond color parameters of the flavored EVOO samples. All the analyses were performed in duplicate and the mean values, along with the standard deviation, were reported.

The free fatty acids (FFAs), peroxide value (PV) and coefficients of extinction (K_232_ and K_270_) are critical parameters used to evaluate the quality of flavored EVOO, mainly in terms of the chemical changes and oil stability. In all the olive oil samples, the FFAs content was much lower than the limit of 0.8% *w*/*w* (expressed as oleic acid) set by EU Regulation 2568/91 for the quality of extra virgin olive oil [39]. The extinction coefficients K_232_ and K_270_ varied significantly between the oil samples flavored with marjoram, at different addition levels and at different storage time. The K_232_ and K_270_ values are indicative of the presence of conjugated dienes and trienes or unsaturated carbonyl compounds, respectively. The slight increase observed in the K_232_ and K_270_ values of the flavored EVOO samples compared to the unflavored EVOO could possibly be attributed to the presence of oxygen in contact with the samples during the ultrasound application, in accordance with Sicaire et al. (2016) [40]. The peroxide value (PV) denotes the content of hydroperoxides produced at the early stages of lipid oxidation. The unflavored EVOO had a higher value for the PV (although not significantly different) compared to the respective ones for the flavored EVOO samples, suggesting that oxidation proceeds at a lower rate in flavored olive oil samples, possibly due to the presence of antioxidant compounds of marjoram [41]. In all cases, all the flavored EVOO samples presented K_232_, K_270_ and PV values below the maximum permitted for virgin olive oil, concerning the quality characteristics of olive oils according to the European legislation (K_232_ ≤ 2.50, K_270_ ≤ 0.22 and PV ≤ 20 meq O_2_/kg) [39]. In line with the above findings, Baiano et al. (2009) [42] carried out a study of the quality indices during the storage of flavored olive oils and found that olive oils flavored with garlic maintained their indices below the maximum level permitted for extra virgin olive oils.

Regarding the oxidative stability measured by Rancimat, the flavoring of EVOO with *Origanum majorana* L. significantly increased the oxidation induction time in comparison to the unflavored oil. In a similar vein, Gambacorta et al. (2007) [43] observed that the addition of various concentrations of garlic, hot pepper, oregano and rosemary improved the long-term stability of olive oils. A number of researchers analyzed the changes in the oxidative status of flavored olive oils to confirm the efficacy of the flavorings’ bioactive properties and the role they played in promoting olive oils’ stability. Ayadi et al. (2009) [11] found that aromatics such as rosemary and thyme protected the olive oil from thermal oxidation. In contrast, Issaoui et al. (2011) [44] noted that lemon and thyme at high concentrations (80 g/kg of oil) did not protect the olive oils from thermo-oxidative processes at the smoking point.

Chlorophylls and carotenoids are pigments contributing to the oxidative stability and to the color of olive oils, ranging from yellow and green to greenish gold [45]. The levels of chlorophyll, measured by the Lovibond Spectrocolorimeter, were significantly higher in the olive oil flavored with 40 g L^−1^ dry marjoram after 14 days of storage compared to the unflavored oil, whereas the b-carotene content increased significantly after 14 days of storage at both flavoring addition levels (Table 1). Ayadi et al. (2009) [11] investigated the changes in the chlorophyll and b-carotene pigments during the thermal oxidation of flavored olive oils and found that those with high levels of chlorophyll and carotenoids demonstrate strong resistance to thermal oxidation. This resistance in flavored olive oils may be attributed to the effectiveness and stability of certain compounds that migrate from aromatic plants to the olive oil during the maceration process.

### 3.2. Evaluation of Antifungal Activity

Figure 1 presents the cell counts of *Zygosaccharomyces bailli* in two batches of mayonnaise (MY B_0 and MY C), both incubated at 25 °C. In the mayonnaise control samples (MY C) prepared with the unflavored EVOO, the *Zygosaccharomyces bailli* cell counts had a steady upward trend, and during the incubation period of 20 days, they increased by approximately 1 log cycle compared to the initial inoculum. In contrast, the mayonnaise samples prepared with the EVOO flavored with 40 g L^−1^ dry marjoram (MY B_0) showed no yeast growth for two weeks. Yeast colonies appeared at the beginning of the third week of incubation at 25 °C. Some of the polyphenolic compounds extracted from the marjoram dry leaves in the olive oil may be the cause of the partial inhibition of *Zygosaccharomyces bailli* cell growth in the mayonnaise prepared with the olive oil flavored with 40 g L^−1^ dry marjoram (MY B_0). According to Rojo et al. (2015) [46], p-coumaric acid has shown limited inhibitory activity (15%) against *Zygosaccharomyces bailli*. It has also been reported that luteolin exhibits activity against *Zygosaccharomyces bailli* and *Zygosaccharomyces rouxii* [47]. The pH value of the samples of both mayonnaise batches was 4.0 during the incubation period of 20 days. The evaluation of the antifungal activity of the flavored EVOO could be applied to the preparation of various foods, such as bakery products, spreadable fats, sauces, condiments and confectionaries, and enhance their microbiological shelf life by partially replacing synthetic preservatives.

### 3.3. Analysis of Volatile Compounds

Olive oil’s distinctive flavor is a result of the VCs produced by the lipoxygenase (LOX) and hydroperoxide homolytic cleavage (13-LOOH) pathways, which also contribute to its quality [48]. Table 2 indicates that the most prevalent compound was (*E*)-2-hexenal, while fourteen VC were identified in the unflavored olive oil (C_0 and C_14) (Figure 2).

Previous research studies [32,49,50] have confirmed the higher abundance of aldehydes in high-quality olive oil, which is mostly due to the presence of (*E*)-2-hexenal, an aldehyde produced via the LOX pathway that contributes to the classic “green note” flavor of olive oil [51]. Other aldehydes detected at lower concentrations are (*Z*)-3-hexenal and nonanal. The latter is produced from the autoxidation reactions occurring after olive oil extraction, whereas (*Z*)-3-hexenal is produced through the LOX pathway [48,52]. The alcohols detected in the unflavored EVOO were C6 compounds (3-hexen-1-ol, (*E*)-2-hexen-1-ol, 1-hexanol) that contribute to the “green” aroma and astringent flavor of olive oil [53,54]. The esters hexyl acetate and (*Z*)-3-hexenyl acetate were also detected, both of which contribute significantly to the flavor of olive oil. Hexyl acetate imparts a fruity, sweet flavor, while (*Z*)-3-hexenyl acetate provides a banana flavor [51,55]. Extra virgin olive oil was flavored with the addition of 20 g L^−1^ and 40 g L^−1^ dry marjoram. The volatile profile of the flavored EVOO was examined after the addition of dry marjoram and the ultrasound (US) application (samples A_0 and B_0), and after storage in a dark place for 14 days (samples A_14 and B_14). The presence of 11 terpene compounds that migrated from dry marjoram was observed in the flavored EVOO, including hydrocarbon monoterpenes (β-thujene, a-pinene, camphene, sabinene, β-pinene, γ-terpinene, terpinolene) and oxygenated monoterpenes (eucalyptol, (−)-terpinen-4-ol, α-terpineol, (+)-camphor). These compounds have previously been identified in *Origanum majorana* L. [56]. Terpenes have beneficial biological features, such as analgesic and anticonvulsant effects. Several studies have shown that some terpenes can lessen inflammatory symptoms [57].

Eucalyptol was the most abundant terpene in the EVOO sample flavored with 20 g L^−1^ dry marjoram (3.76 ± 0.76 mg kg^−1^), followed by sabinene (2.87 ± 0.11 mg kg^−1^). A great presence of terpenes was observed in the olive oil flavored with 40 g L^−1^ dry marjoram, such as a-pinene (3.84 ± 1.14 mg kg^−1^), camphene (3.47 ± 1.17 mg kg^−1^), eucalyptol (3.42 ± 2.85 mg kg^−1^) and sabinene (3.19 ± 0.51 mg kg^−1^). The increase in the quantity of dry marjoram from 20 g L^−1^ (sample A_0) to 40 g L^−1^ (sample B_0) resulted in an increase in the levels of a-pinene, camphene, and (+)-camphor. Also, an increase in the concentration of terpenes occurred after 14 days of storage (samples A_14 and B_14). In contrast, the levels of VCs produced from the LOX pathway (mainly C6 aldehydes, alcohols, and esters) were not significantly affected, which indicates that the use of ultrasound aided the process of flavoring EVOO without altering its flavor components.

### 3.4. Photometric Analysis

Table 3 presents the total phenolic content (TPC), as well as the antiradical and the antioxidant activities, of the flavored EVOO samples. The TPC values of the EVOO samples flavored with marjoram were significantly higher compared to the unflavored EVOO and ranged from 179.18 to 249.84 mg GAE/kg oil, depending on the amount of marjoram added and on the storage time. The antiradical activity of the flavored EVOO oils, expressed as mg Trolox equivalents (TE) per kg of oil, was not significantly affected by the addition of marjoram compared to the unflavored EVOO; however, the antiradical activity of the EVOO flavored with 40 g L^−1^ dry marjoram decreased after 14 days of storage. Similarly, the incorporation of marjoram into the EVOO did not have a significant impact on the antioxidant activity measured by ferric-reducing antioxidant power (FRAP) assay compared to the unflavored EVOO; nevertheless, the antioxidant activity of the flavored EVOO samples at both addition levels decreased after 14 days of storage.

Some studies report conflicting results for the impact of flavoring agents on the total phenolic content, no matter if they belong to the same family. More specifically, for *Lamiaceae* plants, the application of dry basil 15% by means of ultrasound technology [10], dry lavender 5% *w*/*w* and dry mint 5% [11], resulted in a decrease in the phenolic compounds content, whereas the addition of dry oregano 10 g/kg, dry thyme 10 g/kg [58] and dry rosemary 10% *w*/*w* [10] caused an increase in the total phenolic content. The above contradictory findings could be attributed to a series of factors, including the flavoring technique and its conditions, as well as the type, the quantity and the chemical composition of the flavoring agent [13]. Concerning the antioxidant activity, Baiano et al. (2009) [42] studied the use of some flavoring agents (oregano, pepper, lemon, rosemary and garlic) and concluded that the antioxidant content was higher in the flavored than in the non-flavored olive oils, and it was more pronounced after olive oil storage.

### 3.5. Identification of Phenolic Compounds by LC-QToF-MS

A total of 21 compounds were tentatively characterized in the EVOO samples, including phenolic alcohols, phenolic acids, flavonoids, lignans and secoiridoids (Table 4).

The phenolic acids characterized were vanillic acid and p-coumaric acid, while the flavonoids identified were luteolin, naringenin, apigenin, and hesperetin. Hesperetin and p-coumaric acid were only identified in the flavored olive oil, suggesting that their presence is due to the addition of *Origanum majorana* L. These compounds have previously been identified in the leaves of the plant [59].

p-coumaric acid is a nutraceutical compound and a strong inhibitor of oxidative degradation, which enhances the stability of olive oil [60,61]. The CH=CH–COOH group of p-coumaric acid contributes to promoting antioxidant activity and preventing the process of primary oxidation [62,63,64]. The high bioavailability and bioaccessibility of p-coumaric acid promotes the nutritional value of olive oil [61,65] by preventing and treating cardiovascular diseases, inflammation, diabetes, and nervous system conditions, while also demonstrating various pharmacological effects such as anti-tumor, antibacterial, and anti-aging properties [66]. Furthermore, other functions of p-coumaric acid include anxiolytic, analgesic, anti-ulcer, antipyretic, and antiplatelet aggregation activities [65]. Hesperetin is a trihydroxyflavanone, which is considered a powerful antioxidant with high nutritional value. The addition of hesperetin to soybean oil-in-water emulsions has been found to improve the oxidation stability [67]. The supplementation of hesperetin in feed additives improves the lipid-related characteristics of eggs [68]. Hesperetin also exhibits anti-inflammatory, antiviral, and neuroprotective activities [69] and has been shown to inhibit chemically induced mammary tumorigenesis, colon carcinogenesis, reduced heart attack incidents, and blood pressure [70].

Luteolin and apigenin are major flavonoids in olive oils, which are derived from the glucosides found in the drupe [71]. Flavonoids are important for human health as they present anti-inflammatory and antioxidative properties [72]. Hydroxytyrosol and tyrosol are important phenolic compounds of olive oil with antiatherogenic, cardioprotective, anticancer, neuroprotective and endocrine effects [73]. Regarding secoiridoids, the main identified compounds were 3,4-DHPEA-EDA (2-(3,4-hydroxyphenyl) ethyl (3*S*,4*E*)-4-formyl-3-(2-oxoethyl) hex-4-enoate), oleocanthal, and ten isomers of oleuropein aglycone, which have been previously reported in olive oil [74].

3,4-DHPEA-EDA influences the bitter taste of olive oil [75] and displays antioxidant [76], anti-inflammatory [77], and anti-tumor properties [78]. 1-acetoxypinoresinol, a lignan specifically observed in olives was also identified [79]. Lignans are plant polyphenols renowned for their health benefits. The similarity between the chemical structures of lignans and estrogens suggests that they may act as hormonal modulators in breast malignancies via estrogenic or antiestrogenic activity [80,81].

## 4. Conclusions

Olive oil flavoring is an emerging commercial trend, enhancing its sensory and antioxidant profile, as well as the oil’s shelf life. In the present study, *Origanum majorana* L. was incorporated as a flavoring agent in extra virgin olive oil (EVOO), at concentrations of 20 g L^−1^ and 40 g L^−1^, with the aid of ultrasound technology. The addition of marjoram to the EVOO increased the oil’s oxidative stability, as well as the chlorophyll, b-carotene and total phenolic contents; nevertheless, the flavoring did not have a significant impact on the quality attributes, nor on the antioxidant and antiradical activities of the flavored oil. Moreover, the volatile profile of the flavored EVOO was upgraded by the presence of 11 terpene compounds, including hydrocarbon and oxygenated monoterpenes, whereas the LC-QToF-MS analysis revealed the presence of hesperetin and p-coumaric acid in only the flavored EVOO. Finally, the assessment of the antifungal activity of the flavored EVOO in a typical o/w emulsion with *Zygosaccharomyces bailii* as the target microorganism proved that the flavored EVOO could act in synergy and partially replace synthetic antifungals such as benzoic and sorbic acid, commonly used in edible emulsions.

## Figures and Tables

**Figure 1 foods-13-03164-f001:**
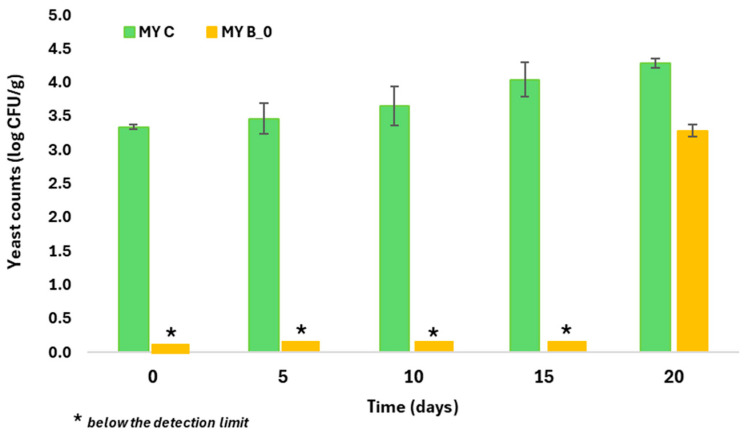
Yeast counts of *Zygosaccharomyces bailli* in samples of two batches of mayonnaise over an incubation period of 20 days at 25 °C. Yeast colonies were measured on YPD agar plates incubated at 25 °C for 48 h. MY B_0: mayonnaise samples prepared with EVOO flavored with 40 g L^−1^ dry marjoram; MY C: mayonnaise samples prepared with the unflavored EVOO.

**Figure 2 foods-13-03164-f002:**
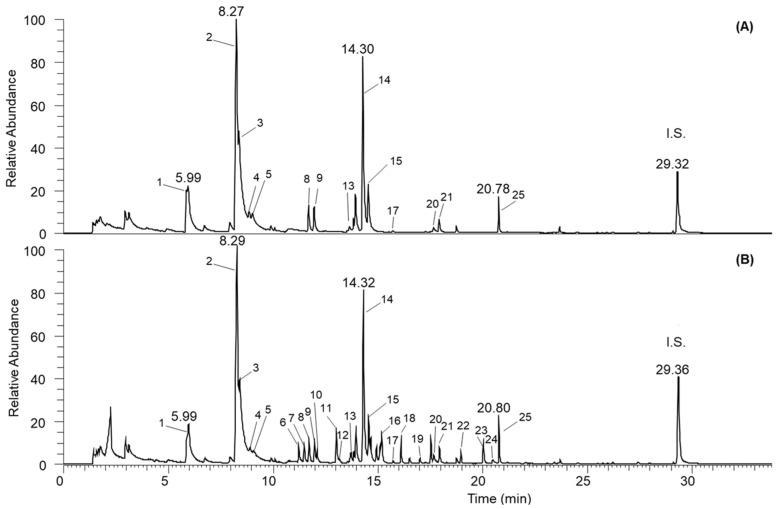
Chromatograms of (**A**) EVOO and (**Β**) EVOO flavored with 40 g L^−1^ dry marjoram. Peak numbers correspond to compounds in Table 2 (I.S.: internal standard).

**Table 1 foods-13-03164-t001:** Quality indices, oxidation induction time (OIT) and Lovibond color parameters of the flavored EVOO samples.

Quality Indices	EVOO Samples				
	C_0	A_0	B_0	A_14	B_14
Free Fatty Acids (FFA, % *w*/*w* oleic acid)	0.56 ± 0.00 a	0.56 ± 0.00 a	0.55 ± 0.01 a	0.50 ± 0.07 a	0.59 ± 0.08 a
K_232_	1.26 ± 0.03 a	1.78 ± 0.01 d	1.45 ± 0.04 c	2.25 ± 0.02 e	1.37 ± 0.02 b
K_270_	0.10 ± 0.01 a	0.09 ± 0.01 a	0.18 ± 0.01 c	0.15 ± 0.01 b	0.22 ± 0.01 d
ΔΚ	0.011 ± 0.001 a	0.011 ± 0.001 a	0.011 ± 0.001 a	0.010 ± 0.001 a	0.016 ± 0.001 b
Peroxide Value (PV, meq O_2_/kg)	15.35 ± 1.30 a	12.50 ± 2.32 a	11.86 ± 1.79 a	14.86 ± 2.65 a	13.88 ± 2.74 a
Oxidation Induction Time (OIT, h)	11.16 ± 0.02 a	11.67 ± 0.07 b	12.70 ± 0.23 d	11.94 ± 0.12 c	13.34 ± 0.18 e
**Lovibond color parameters**					
R (Red)	1.0	1.2	1.3	1.3	1.2
Y (Yellow)	35.5	35	35.5	36	38
B (Blue)	0	0	0	0	0
N (Neutral)	0.3	0.3	0.8	0.6	0.5
Chlorophyll (ppm)	6.84 ± 0.02 a	6.86 ± 0.00 a	6.92 ± 0.04 b	6.86 ± 0.01 a	7.31 ± 0.06 c
b-carotene (ppm)	44.85 ± 0.01 a	44.86 ± 0.01 a	45.17 ± 0.03 c	45.04 ± 0.02 b	46.33 ± 0.10 d

C_0: unflavored EVOO; A_0: EVOO flavored with 20 g L^−1^ dry marjoram; B_0: EVOO flavored with 40 g L^−1^ dry marjoram; A_14: EVOO flavored with 20 g L^−1^ dry marjoram after 14 days of storage; B_14: EVOO flavored with 40 g L^−1^ dry marjoram after 14 days of storage. Mean values bearing different lowercase letters in the same row differ significantly (*p* < 0.05). (a < b < c < d < e).

**Table 2 foods-13-03164-t002:** Volatile compounds identified and semi-quantified (mg kg^−1^) in the EVOO samples.

Peak No	Compound	^1^ RI Exp.	^2^ RI Lit.	EVOO Samples					
				C_0 ^3^	C_14 ^4^	A_0 ^5^	A_14 ^6^	B_0 ^7^	B_14 ^8^
1	(*Z*)-3-hexenal	<800	797	6.33 ± 2.01 a	4.88 ± 0.43 a	7.91 ± 0.30 a	4.87 ± 0.10 a	7.14 ± 1.50 a	3.89 ± 0.90 a
2	(*E*)-2-hexenal	849	846	49.51 ± 13.31 a	64.92 ± 0.98 a	70.06 ± 5.94 a	72.79 ± 1.58 a	75.87 ± 15.49 a	62.25 ± 6.40 a
3	3-hexen-1-ol	852	847	42.45 ± 16.28 a	55.35 ± 4.63 a	58.36 ± 4.70 a	52.09 ± 2.40 a	59.41 ± 15.20 a	44.25 ± 6.49 a
4	(*E*)-2-hexen-1-ol	864	851	5.96 ± 1.82 a	6.84 ± 1.03 a	7.91 ± 0.62 a	6.80 ± 0.56 a	7.35 ± 1.62 a	6.03 ± 1.14 a
5	1-hexanol	868	863	10.44 ± 4.88 a	13.47 ± 0.83 a	14.33 ± 1.34 a	11.78 ± 1.85 a	12.54 ± 2.71 a	10.22 ± 0.84 a
6	β-thujene	923	920	−	−	2.72 ± 0.42 ab	4.22 ± 0.31 b	2.46 ± 0.87 ac	3.34 ± 0.14 ab
7	a-pinene	929	932	−	−	1.67 ± 0.24 a	2.56 ± 0.27 ab	3.84 ± 1.14 c	3.38 ± 0.06 a
8	3-Ethyl-1,5-octadiene isomer 1	936	930	3.83 ± 1.22 a	4.61 ± 0.59 a	5.23 ± 0.53 a	4.57 ± 0.24 a	5.04 ± 1.59 a	3.93 ± 0.30 a
9	3-Ethyl-1,5-octadiene isomer 2	943	930	4.12 ± 1.33 a	4.90 ± 0.63 a	4.80 ± 0.40 a	4.03 ± 0.10 a	4.35 ± 1.15 a	3.36 ± 0.27 a
10	Camphene	946	946	−	−	1.54 ± 0.03 a	1.83 ± 0.11 ab	3.47 ± 1.17 b	2.51 ± 0.15 ab
11	Sabinene	970	969	−	−	2.87 ± 0.11 a	6.23 ± 0.18 b	3.19 ± 0.51 a	5.67 ± 0.50 bc
12	β-pinene	974	974	−	−	0.24 ± 0.03 a	0.32 ± 0.05 a	0.29 ± 0.40 a	0.37 ± 0.11 a
13	6-methyl-5-hepten-2-one	984	989	0.22 ± 0.01 a	0.25 ± 0.02 a	0.17 ± 0.03 a	0.30 ± 0.09 a	0.25 ± 0.11 a	0.33 ± 0.02 a
14	(*Z*)-3-hexenyl acetate	1004	1004	30.38 ± 8.32 a	34.37 ± 2.52 a	39.65 ± 5.12 a	35.12 ± 2.01 a	35.62 ± 10.67 a	32.65 ± 3.07 a
15	Hexyl acetate	1012	1007	10.28 ± 3.09 a	12.82 ± 1.05 a	10.32 ± 1.55 a	10.15 ± 3.04 a	9.07 ± 0.11 a	7.22 ± 0.72 a
16	Eucalyptol	1028	1026	−	−	3.76 ± 0.76 a	3.89 ± 0.79 a	3.42 ± 2.85 a	5.08 ± 0.14 a
17	(*E*)-β-ocimene	1047	1044	0.26 ± 0.05 a	0.42 ± 0.04 a	0.40 ± 0.05 a	0.36 ± 0.00 a	0.45 ± 0.11 a	0.36 ± 0.02 a
18	γ-terpinene	1085	1086	−	−	2.86 ± 0.32 ab	4.52 ± 0.03 c	2.73 ± 0.78 a	3.67 ± 0.28 abc
19	Terpinolene	1085	1086	−	−	0.69 ± 0.00 a	1.00 ± 0.01 a	0.68 ± 0.21 a	0.79 ± 0.00 a
20	Nonanal	1105	1100	1.35 ± 0.03 a	1.70 ± 0.06 a	1.98 ± 0.13 a	1.76 ± 0.01 a	2.03 ± 1.12 a	2.15 ± 0.06 a
21	2-ethenyl-1.1-dimethyl-3-methylenecyclohexane	1114	−	2.43 ± 0.40 a	2.92 ± 0.19 a	3.26 ± 0.05 a	2.60 ± 0.02 a	3.19 ± 1.30 a	2.81 ± 0.12 a
22	(+)-camphor	1146	1141	−	−	0.71 ± 0.01 ab	0.77 ± 0.01 a	1.98 ± 0.60 c	1.58 ± 0.02 abc
23	(-)-terpinen-4-ol	1181	1174	−	−	1.52 ± 0.03 a	2.76 ± 0.03 b	1.71 ± 0.60 a	3.69 ± 0.17 c
24	α-terpineol	1196	1186	−	−	0.37 ± 0.07 a	0.58 ± 0.03 a	0.50 ± 0.30 a	1.14 ± 0.06 b
25	Methylcyclodecane	1205	1202	4.42 ± 0.95 a	5.11 ± 0.25 a	5.74 ± 0.25 a	4.19 ± 0.11 a	6.43 ± 2.05 a	5.73 ± 0.33 a

^1^ RI exp. = experimental retention index; ^2^ RI lit. = literature retention index; ^3^ C_0: unflavored EVOO at 0 day of storage; ^4^ C_14: unflavored EVOO after 14 days of storage; ^5^ A_0: EVOO flavored with 20 g L^−1^ dry marjoram; ^6^ A__14: EVOO flavored with 20 g L^−1^ dry marjoram after 14 days of storage; ^7^ B_0: EVOO flavored with 40 g L^−1^ dry marjoram; ^8^ B_14: EVOO flavored with 40 g L^−1^ dry marjoram after 14 days of storage. Mean values with different letters in the same row are significantly different (*p* < 0.05).

**Table 3 foods-13-03164-t003:** Total phenolic content and antiradical and antioxidant activities of the EVOO samples.

Olive Oil Samples	Total Phenolic Content (TPC) Expressed as mg Gallic Acid Equivalents (GAE)/kg Oil	Antiradical Activity Expressed as mg Trolox Equivalents (TE)/kg Oil	Antioxidant Activity by Ferric Reducing Antioxidant Power (FRAP) Assay Expressed as mg of FeSO_4_ x 7H_2_O/kg Oil
C_0	117.75 ± 17.10 a	204.81 ± 13.54 ab	1354.74 ± 99.71 ab
A_0	179.18 ± 11.43 b	192.42 ± 14.42 a	1581.28 ± 136.20 b
B_0	215.98 ± 7.63 cd	238.74 ± 14.05 b	1402.36 ± 98.43 ab
A_14	183.80 ± 17.30 bc	205.10 ± 17.12 ab	1420.26 ± 97.31 ab
B_14	249.84 ± 8.02 d	199.17 ± 7.02 a	1236.74 ± 50.54 a

C_0: unflavored EVOO; A_0: EVOO flavored with 20 g L^−1^ dry marjoram; B_0: EVOO flavored with 40 g L^−1^ dry marjoram; A_14: EVOO flavored with 20 g L^−1^ dry marjoram after 14 days of storage; B_14: EVOO flavored with 40 g L^−1^ dry marjoram after 14 days of storage. Mean values bearing different lowercase letters in the same column differ significantly (*p* < 0.05). (a < b < c < d).

**Table 4 foods-13-03164-t004:** Phenolic compounds identified in the olive oil samples by LC-QToF-MS.

Compound	Sample	t_R_ (min)	Formula [M-H]^−^	Theoretical *m*/*z* [M-H]^−^	Experimental *m*/*z* [M-H]^−^	Mass Error
Hydroxytyrosol	C_0, A_14, B_14	1.88	C_8_H_10_O_3_	153.0557	153.0553	2.73
Tyrosol	C_0, A_14, B_14	2.76	C_8_H_10_O_2_	137.0608	137.0606	1.49
Vanillic acid	C_0, A_14, B_14	3.67	C_8_H_8_O_4_	167.0350	167.0350	0.00
p-coumaric acid	A_14, B_14	5.30	C_9_H_8_O_3_	163.0400	163.0404	−2.39
3,4-DHPEA-EDA	C_0, A_14, B_14	7.46	C_17_H_20_O_6_	319.1187	319.1188	−0.27
Oleuropein aglycon isomer 1	C_0, A_14, B_14	7.90	C_19_H_22_O_8_	377.1242	377.1229	3.43
Oleuropein aglycon isomer 2	C_0, A_14, B_14	8.30	C_19_H_22_O_8_	377.1242	377.1232	2.63
Oleocanthal	C_0, A_14, B_14	9.15	C_17_H_20_O_5_	303.1238	303.1238	0.00
Oleuropein aglycon isomer 3	C_0, A_14, B_14	9.42	C_19_H_22_O_8_	377.1242	377.1235	1.83
Luteolin	C_0, A_14, B_14	9.86	C_15_H_10_O_6_	285.0405	285.0405	0.00
Oleuropein aglycon isomer 4	C_0, A_14, B_14	9.96	C_19_H_22_O_8_	377.1242	377.1228	3.69
1-Acetoxypinoresinol	C_0, A_14, B_14	10.09	C_22_H_24_O_8_	415.1398	415.1386	2.99
Oleuropein aglycon isomer 5	C_0, A_14, B_14	10.33	C_19_H_22_O_8_	377.1242	377.1228	3.69
Oleuropein aglycon isomer 6	C_0, A_14, B_14	10.57	C_19_H_22_O_8_	377.1242	377.1231	2.89
Oleuropein aglycon isomer 7	C_0, A_14, B_14	10.77	C_19_H_22_O_8_	377.1242	377.1232	2.63
Naringenin	C_0, A_14, B_14	10.94	C_15_H_12_O_5_	271.0612	271.0611	0.37
Apigenin	C_0, A_14, B_14	11.48	C_15_H_10_O_5_	269.0456	269.0455	0.19
Oleuropein aglycon isomer 8	C_0, A_14, B_14	11.68	C_19_H_22_O_8_	377.1242	377.1234	2.10
Oleuropein aglycon isomer 9	C_0, A_14, B_14	11.92	C_19_H_22_O_8_	377.1242	377.1234	2.10
Oleuropein aglycon isomer 10	C_0, A_14, B_14	12.29	C_19_H_22_O_8_	377.1242	377.1234	2.10
Hesperetin	A_14, B_14	12.63	C_16_H_14_O_6_	301.0718	301.0714	1.20

t_R_: retention time; C_0: unflavored EVOO; A_14: EVOO flavored with 20 g L^−1^ dry marjoram after 14 days of storage; B_14: EVOO flavored with 40 g L^−1^ dry marjoram after 14 days of storage.

## Data Availability

The original contributions presented in the study are included in the article, further inquiries can be directed to the corresponding author.

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
