# Peer review of "Origanum majorana L. As Flavoring Agent: Impact on Quality Indices, Stability, and Volatile and Phenolic Profiles of Extra Virgin Olive Oil (EVOO)"

_foods, 2024, doi:10.3390/foods13193164_

Round 1

Reviewer 1 Report

Comments and Suggestions for Authors

Manuscript entitled Origanum majorana L. as flavoring agent: Impact on quality indices, stability, volatile and phenolic profile of extra virgin olive oil (EVOO) addresses the application of Origanum majorana L. as a flavoring agent of extra virgin olive oil with the aim to assess the impact of flavoring on its quality indices, oxidative stability, antioxidant, antiradical and antifungal activities, and volatile and phenolic profile of oil.

The manuscript is scientifically sound, composed according the scientific methodology for writing the scientific papers and represents a valuable contribution for scientists involved in the phytochemistry and related fields.

In the introduction section authors concisely provided the background information of the study topic and explained the principal aims of the study. Materials and Methods section, provided sufficient data in relation to the study material, analytical techniques used and statistical methods for data analysis.

The results of study are presented in tabulated and graphical form, in a proper format and easy to understand and interpret. In addition a chromatograms of identified oil volatile compounds are presented.  Throughout this section, results are adequately discussed and authors cited sufficient references of recent studies to support and explain the obtained data.

The conclusion section summarized the results of this study, indicating the most significant findings and their importance. References are properly cited.

Suggestions:

Please indicate in which form the plant material was introduced to olive oil, and how it was stored before the use?

Minor language corrections needed.

Comments on the Quality of English Language

Minor language corrections needed.

Author Response

Point to point answers to Reviewer’s 1 comments

Manuscript entitled Origanum majorana L. as flavoring agent: Impact on quality indices, stability, volatile and phenolic profile of extra virgin olive oil (EVOO) addresses the application of Origanum majorana L. as a flavoring agent of extra virgin olive oil with the aim to assess the impact of flavoring on its quality indices, oxidative stability, antioxidant, antiradical and antifungal activities, and volatile and phenolic profile of oil.

The manuscript is scientifically sound, composed according the scientific methodology for writing the scientific papers and represents a valuable contribution for scientists involved in the phytochemistry and related fields.

In the introduction section authors concisely provided the background information of the study topic and explained the principal aims of the study. Materials and Methods section, provided sufficient data in relation to the study material, analytical techniques used and statistical methods for data analysis.

The results of study are presented in tabulated and graphical form, in a proper format and easy to understand and interpret. In addition a chromatograms of identified oil volatile compounds are presented.  Throughout this section, results are adequately discussed and authors cited sufficient references of recent studies to support and explain the obtained data.

The conclusion section summarized the results of this study, indicating the most significant findings and their importance. References are properly cited.

Suggestions:

  1. Please indicate in which form the plant material was introduced to olive oil, and how it was stored before the use?

Response: We indeed appreciate and thank the Reviewer for his/her nice comments. Regarding the form of the plant material, the marjoram leaves were dried in a laboratory oven at 40°C prior to their addition to olive oil. The aromatic plant was stored in a dark and dry place at room temperature before use.

Reviewer 2 Report

Comments and Suggestions for Authors

The article addresses the use of Origanum majorana L. as a flavoring agent in extra-virgin olive oil, evaluating its impact on quality indices, oxidative stability, phenolic and volatile profiles, as well as the antifungal activity of the product, using ultrasound-assisted maceration. Although the use of plants from the Lamiaceae family has been explored in other works, the focus on Origanum majorana L. and its interaction with olive oil presents a contribution, particularly regarding antifungal activity. Unfortunately, the addition of Origanum majorana L. did not have a significant effect on the antiradical and antioxidant activity of the flavored oil, which reduces the impact of the results.

 Several adjustments can be made to improve the quality of the manuscript:

 -          Justify the choice of extraction conditions (time and temperature), as well as the amount of marjoram added (20 and 40g/L) and the storage conditions.

-          Although the use of ultrasound is innovative, it would be important to compare its performance with other extraction methods, such as traditional maceration or microwave extraction. A comparison with other techniques could strengthen the relevance of choosing ultrasound and highlight its advantages.

-          Explain why only sample B (MYB_0) was chosen for antifungal activity evaluation, and why sample A was not evaluated. It is also unclear why the samples were not tested after 14 days of storage. Please justify.

-          The antifungal activity was only partial, with inhibition of Zygosaccharomyces bailli for just two weeks, which limits its practical application in longer storage systems.

-          The exclusive use of mayonnaise as a model system to evaluate antifungal activity restricts the applicability of the results. Why was the effectiveness not tested in a wider range of fungus-susceptible foods?

-          Figure 1 needs improvement to better visualize the absence of growth in sample MYB_0 for up to 14 days.

-          Expand the discussions, for example, regarding the absence of significant effects on antioxidant and antiradical activity, and further explore the applicability of the antifungal results in different foods.

-          Although the authors identified an increase in phenolic compounds, the discussion on the impact of these compounds on olive oil quality and their health benefits is limited. A deeper analysis of how these compounds influence the nutritional value and stability of the oil could add more relevance.

 Therefore, I consider that the manuscript, as currently presented, is not suitable for publication and requires major revisions and re-evaluation.

Author Response

Point to point answers to Reviewer’s 2 comments

The article addresses the use of Origanum majorana L. as a flavoring agent in extra-virgin olive oil, evaluating its impact on quality indices, oxidative stability, phenolic and volatile profiles, as well as the antifungal activity of the product, using ultrasound-assisted maceration. Although the use of plants from the Lamiaceae family has been explored in other works, the focus on Origanum majorana L. and its interaction with olive oil presents a contribution, particularly regarding antifungal activity. Unfortunately, the addition of Origanum majorana L. did not have a significant effect on the antiradical and antioxidant activity of the flavored oil, which reduces the impact of the results.

Several adjustments can be made to improve the quality of the manuscript:     

  1. Justify the choice of extraction conditions (time and temperature), as well as the amount of marjoram added (20 and 40g/L) and the storage conditions.

Response: Thank you for your suggestion. The ultrasound extraction temperature and time were selected based on previous experiments and on literature review. Long-time exposure to ultrasounds normally results in loss or degradation of compounds, whereas high temperatures may induce lipid oxidation, loss of volatiles and degradation of bioactive compounds. Concerning the amount of plant material and the storage conditions applied, these were chosen after studying previous works related to flavoring of olive oils with other Lamiaceae plants and for comparative reasons.

  1. Although the use of ultrasound is innovative, it would be important to compare its performance with other extraction methods, such as traditional maceration or microwave extraction. A comparison with other techniques could strengthen the relevance of choosing ultrasound and highlight its advantages.

Response: We thank the Reviewer for his/her useful suggestion; however, the comparison and the assessment of the different flavoring techniques of oils (i.e. co-extraction, direct contact with the aid of ultrasound, microwave, or heating, or traditional maceration, etc.) was out of the scope of the present study. It could be an interesting research topic to investigate and could be the subject of a future study.

  1. Explain why only sample B (MYB_0) was chosen for antifungal activity evaluation, and why sample A was not evaluated. It is also unclear why the samples were not tested after 14 days of storage. Please justify.

Response: Thank you for this remark. The scope was to perform a preliminary assessment of the antifungal activity and not to analyze the minimum inhibitory concentration of the dry plant material, which would have required a much more analytical microbiological study. Therefore, we assessed only the maximum concentration in order to detect the presence of any antifungal activity. Regarding the storage time, we would like to point out that, according to the methodology described in paragraph 2.5.2, the total storage period for microbiological analysis was 20 days (Line 160 and Figure 1).

  1. The antifungal activity was only partial, with inhibition of Zygosaccharomyces bailli for just two weeks, which limits its practical application in longer storage systems.

Response: The aim of the challenge test was to investigate the potential antifungal activity of the flavored olive oil on real food. Of course, further research is needed to establish this flavored olive oil as a substitute of the conventional preservatives like sorbates and benzoates. Besides, we have to take on account the fact that the specific food (mayonnaise) was not packaged and it was tested in conditions that simulate its use in catering facilities in bulk containers that are not sealed.

  1. The exclusive use of mayonnaise as a model system to evaluate antifungal activity restricts the applicability of the results. Why was the effectiveness not tested in a wider range of fungus-susceptible foods?

Response: Mayonnaise was used as a model food system because: (a) mayonnaise is a widely distributed food in which conventional preservatives are used and (b) it is an emulsion of water-in-oil. Regarding the second question, Zygosaccharomyces bailli was selected because it is one of the dominant spoilage yeasts commonly found in foods. However, following the Reviewer suggestion, a future research area would be the use of the flavored olive oil in bakery products like toast bread, condiments, and in foods, where the antifungal activity can act in combination with other microbial hurdles.

  1. Figure 1 needs improvement to better visualize the absence of growth in sample MYB_0 for up to 14 days.

Response: Figure 1 was revised, according to Reviewer’s suggestion.

  1. Expand the discussions, for example, regarding the absence of significant effects on antioxidant and antiradical activity, and further explore the applicability of the antifungal results in different foods.

Response:  The absence of significant effects on antioxidant and antiradical activity of flavored EVOO, in the current study, could be attributed to a series of factors, such as the flavoring technique and its conditions, as well as the type, the quantity and the chemical composition of flavoring agent (L. 422-425). Olive oil, and all edible oils in general, can be used as carriers of the components of marjoram. Also, all edible oils are used as ingredients in foods such as bakery products, spreadable fats, sauces, condiments and confectionaries. The majority of the above-mentioned foods are characterized by low water activity, inhibiting the growth of most bacteria; however, enabling osmophilic yeasts to grow (such as Zygosaccharomyces bailli) and create spoilage phenomena. Therefore, the preparation of foods with marjoram flavored oils could further enhance their microbiological shelf life. The relative discussion was added in paragraph 3.2 (L. 330-333).

  1. Although the authors identified an increase in phenolic compounds, the discussion on the impact of these compounds on olive oil quality and their health benefits is limited. A deeper analysis of how these compounds influence the nutritional value and stability of the oil could add more relevance.

Therefore, I consider that the manuscript, as currently presented, is not suitable for publication and requires major revisions and re-evaluation.

Response: We thank the reviewer for the valuable comments. A discussion has been added at section “3.5. Identification of phenolic compounds by LC-QToF-MS”.

Reviewer 3 Report

Comments and Suggestions for Authors

The manuscript entitled “Origanum majorana L. as flavoring agent: Impact on quality indices, stability, volatile and phenolic profile of extra virgin olive oil (EVOO)” is a comprehensive study of extra virgin olive oil flavored with Origanum majorana L. A number of well-designed and reported studies have been conducted. The results are clearly described and discussed with the scientific literature. I present some minor comments and suggestions below. At the same time, I suggest paying attention to the journal's editorial requirements and the language aspect.

Comments:

Keywords

flavored EVOO - I do not recommend using shortcuts in Keywords

color - I recommend removing

b-carotene -  b-carotene?? (not only in keywords but in the whole article)

Introduction

Lines 35-38 and 38-41

This is not clear to me; doesn't the second sentence (38-41) undermine the first (35-38)? please rephrase the second sentence

Lines 299-304

These are the results, but what conclusions can be drawn from this? Are there any literature data available for discussion in this area?

Table 2

Table caption and header are not obvious; should be considered...average values ​​of what?

Figure 2

I found no mention of figure 2 in the manuscript text

Line 376 and 376

total phenolic content (TPC) - it is enough to enter the abbreviation once; there is no need to do it again in the next sentence

The manuscript should be intensively checked for language and editing (you should also revisit the Instructions for Authors); sample comments:

Formatting section and subsection titles

Consider consistency when editing tables (Table 1, 2 and 3). Also there is no need to separate the mean and standard deviation (see Table 2; it only makes the data harder to read).

Line 70

“…and ? the impact of flavoring on EVOO quality” - there is no verb here

Line 84

added to …

Line 91

Similarly

Line 102-106

this sentence is too long and therefore difficult to understand

Line 119-121

editing errors

Lines 124, 337, 438

Latin phrases should be italicized

Line 195

Dot is missing

Line 240

-1 in superscript

Line 281

Cases

Lines 356-360

prefix kilo - lower case letter

Etc…

These are just examples of issues that should be paid attention to

Comments on the Quality of English Language

Moderate editing of English language required.

Author Response

Response to Reviewer’s 3 Comments

The manuscript entitled “Origanum majorana L. as flavoring agent: Impact on quality indices, stability, volatile and phenolic profile of extra virgin olive oil (EVOO)” is a comprehensive study of extra virgin olive oil flavored with Origanum majorana L. A number of well-designed and reported studies have been conducted. The results are clearly described and discussed with the scientific literature. I present some minor comments and suggestions below. At the same time, I suggest paying attention to the journal's editorial requirements and the language aspect.

Response: We appreciate the Reviewer for his/her nice comments and we have revised the manuscript according to the proposed suggestions.

Comments:

  1. Keywords

flavored EVOO - I do not recommend using shortcuts in Keywords

color - I recommend removing

b-carotene -  b-carotene?? (not only in keywords but in the whole article)

Response: Keywords were revised according to Reviewer’s suggestion.

  1. Introduction

Lines 35-38 and 38-41

This is not clear to me; doesn't the second sentence (38-41) undermine the first (35-38)? please rephrase the second sentence

Response: The sentences were rephrased according to Reviewer’s suggestion.

  1. Lines 299-304

These are the results, but what conclusions can be drawn from this? Are there any literature data available for discussion in this area?

Response: We thank the Reviewer for the comment. Actually, there are not any literature data concerning the change of chlorophyll and b-carotene pigments after the flavoring of oils with the specific aromatic plant of marjoram. Both pigments are important in olive oil stability. Ayadi et al. (2009) investigated the changes in chlorophyll and b-carotene pigments during the thermal oxidation of flavored olive oils and found that those with high levels of chlorophyll and carotenoids demonstrate strong resistance to thermal oxidation. This resistance in flavored olive oils may be attributed to the effectiveness and stability of certain compounds that migrate from aromatic plants to the olive oil during the maceration process.

The above reference was added in Lines 348-353 of the revised manuscript.

  1. Table 2

Table caption and header are not obvious; should be considered...average values ​​of what?

Response: Table 2 caption and header were revised accordingly.

  1. Figure 2

I found no mention of figure 2 in the manuscript text

Response: We thank the Reviewer for his/her suggestion. Figure 2 was mentioned in paragraph 3.3.

  1. Line 376 and 376

total phenolic content (TPC) - it is enough to enter the abbreviation once; there is no need to do it again in the next sentence

Response: We have revised the abbreviation according to Reviewer’s suggestion.

  1. The manuscript should be intensively checked for language and editing (you should also revisit the Instructions for Authors); sample comments:

Formatting section and subsection titles

Consider consistency when editing tables (Table 1, 2 and 3). Also there is no need to separate the mean and standard deviation (see Table 2; it only makes the data harder to read).

Response: We have checked for language and editing the whole manuscript and we have revised Table 2 according to Reviewer’s suggestion.

  1. Line 70

“…and ? the impact of flavoring on EVOO quality” - there is no verb here

Response: The revision was made according to Reviewer’s suggestion.

  1. Line 84

added to …

Response: The revision was made according to Reviewer’s suggestion.

  1. Line 91

Similarly

Response: The revision was made according to Reviewer’s suggestion.

  1. Line 102-106

this sentence is too long and therefore difficult to understand

Response: The sentence was revised according to Reviewer’s suggestion.

  1. Line 119-121

editing errors

Response: Editing errors were revised according to Reviewer’s suggestion.

  1. Lines 124, 337, 438

Latin phrases should be italicized

Line 195

Dot is missing

Line 240

-1 in superscript

Line 281

Cases

Lines 356-360

prefix kilo - lower case letter

Etc…

These are just examples of issues that should be paid attention to

Response: The above-mentioned issues were revised according to Reviewer’s suggestion.

  1. Comments on the Quality of English Language

Moderate editing of English language required.

Response: English language was thoroughly checked throughout the manuscript.

  1. We notice that the repetition rate is higher than we expected. Please revise
    your manuscript according to the iThenticate report attached and make sure
    that make sure that there is no large part repetition with published papers,
    as well as the total repetition rate is lower than 30% and the single
    repetition rate is lower than 5%.

Response: The manuscript was revised accordingly in order to conform to the total repetition rate of the journal.